# The methyltransferase domain of DNMT1 is an essential domain in acute myeloid leukemia independent of *DNMT3A* mutation

Balpreet Bhogal[1], Barbara A. Weir[1], Ramona Crescenzo[2], Ann Marien[2], Min Chul Kwon[2], Ulrike Philippar[2] & Glenn S. Cowley [1✉]

Aberrant DNA methylation patterns are a prominent feature of cancer. Methylation of DNA is mediated by the DNA methyltransferase (DNMT) protein family, which regulates de novo (*DNMT3A* and *DNMT3B*) and maintenance (*DNMT1*) methylation. Mutations in *DNMT3A* are observed in approximately 22% of acute myeloid leukemia (AML). We hypothesized that *DNMT1* or *DNMT3B* could function as a synthetic lethal therapeutic strategy for *DNMT3A*-mutant AML. CRISPR-Cas9 tiling screens were performed to identify functional domains within *DNMT1/DNMT3B* that exhibited greater dependencies in *DNMT3A* mutant versus wild-type cell lines. Although increased sensitivity to *DNMT1* mutation was observed in some *DNMT3A* mutant cellular models tested, the subtlety of these results prevents us from basing any conclusions on a synthetic lethal relationship between *DNMT1* and *DNMT3A*. Our data suggests that a therapeutic window for DNMT1 methyltransferase inhibition in *DNMT3A*-driven AML may exist, but validation in more biologically relevant models is required.

[1] Janssen Research and Development LLC, Spring House, PA 19477, USA. [2] Janssen Research and Development LLC, Beerse, Belgium.
✉email: gcowley@its.jnj.com

Epigenetic marks impact gene regulation without altering the DNA sequence. The most observed epigenetic mark in the mammalian genome is DNA methylation at the 5-position of cytosines in symmetric CpG dinucleotides. DNA methylation is essential for mammalian development and plays critical roles in several biological processes, including regulation of gene expression, genomic imprinting, X-chromosome inactivation, stem cell regulation, and transposon silencing[1].

DNA methylation of cytosines in CpG dinucleotide pairs is mediated by the DNA methyltransferase (DNMT) protein family. To date, there are five members of the DNMT protein family, of which three are known to exhibit enzymatic activity on mammalian DNA. The initiating DNA methylation mark on cytosines is mediated by the de novo DNA methyltransferases, DNMT3A and DNMT3B[2,3]. Both DNMT3A and DNMT3B comprise Pro-Trp-Trp-Pro (PWWP) and Atrx-Dnmt3-Dnmt3l (ADD) domains that are reported to bind to DNA and interact with histone tails, as well as a C-terminal catalytic methyltransferase (MTase) domain[4]. Once methylation has been established by the de novo methyltransferases, these marks are maintained in subsequent cell divisions by the maintenance DNA methyltransferase, DNMT1[5]. Although DNMT1 contains a MTase domain in the C-terminus of the protein similar to the de novo methyltransferases, it contains functional domains different from DNMT3A and DNMT3B upstream of its catalytic domain.

Previous reports have examined the expression and mutation of DNMT family members in cancer. DNMT1, DNMT3A, and DNMT3B are overexpressed across several types of cancer lineages, including acute myeloid leukemia (AML), melanoma, breast cancer, colorectal cancer, prostate cancer, and stomach cancer[6–11]. Mutations in DNMT3A are observed in patients with myelodysplastic syndromes (MDS, 2–10%) and AML (~22%)[12,13]. Of the DNMT3A mutations observed in AML patients, over half encompass a missense mutation at the R882 amino acid position[13]. DNMT3A mutations are associated with poor prognosis in MDS and AML patients, and increased risk of MDS transitioning to AML[14–17]. Therefore, a therapeutic strategy to target DNMT3A mutant MDS/AML is of high interest.

A previous study identified a putative trigenic negative interaction between DNMT3A, DNMT3B, and DNMT1[18]. We hypothesized that DNMT1 or DNMT3B could be used as a synthetic lethal therapeutic strategy for DNMT3A-mutant AML. CRISPR-Cas9 tiling-single guide RNA (sgRNA) knockout screens have been shown as an efficient screening paradigm to identify essential protein domains within a protein-of-interest[19–21]. Using a custom pooled lentiviral library with sgRNAs tiling the coding region of DNMT1 and DNMT3B, we performed CRISPR-Cas9 knockout tiling screens across a panel of AML cell lines either wild-type or mutant for DNTM3A. While knockout mutations of DNMT3B had minimal effects on cell proliferation across the AML cell lines screened, we identified sgRNAs spanning DNMT1 that resulted in decreased cell proliferation. Independent analyses using STARS[22] and ProTiler[19] identified the catalytic MTase domain of DNMT1 as the most essential functional domain for viability. While DNMT1 knockout mutations did exhibit stronger effects in some AML cell lines screened compared to others, we were unable to determine if DNMT1 is synthetic lethal with DNMT3A mutational status in these cell lines. We engineered paired isogenic cell lines that were either wild-type or mutant for DNMT3A to help minimize background genetic influence to focus on the hypothesized genetic interaction. CRISPR-Cas9 tiling screens across some isogenic cell lines revealed higher sensitivity to DNMT1 knockout mutations in the cell line mutant for DNMT3A compared to the isogenic wild-type cell line, with sgRNAs targeting the MTase domain exhibiting the most consistent results. However, the subtlety in these results prevents us from making conclusions on a synthetic lethal relationship between DNMT1 and DNMT3A. Our studies suggest that a therapeutic window for DNMT1 methyltransferase inhibition in DNMT3A-driven AML may exist, but validation in more biologically relevant models will be required.

## Results

**Knockdown or knockout of DNMT1 affects cell proliferation in AML cell lines independent of DNMT3A mutational status.** Because the de novo DNA methyltransferase DNMT3A is mutated in more than 22% of AML, we wanted to determine if either of the other catalytically active DNMT protein family members, DNMT1 or DNMT3B, can exhibit a synthetic lethal relationship with DNMT3A mutation. The Cancer Dependency Map (DepMap) provides publicly available data from genome-wide pooled shRNA and CRISPR screens across a large panel of cancer cell lines[23,24]. We first looked at the effects of shRNA knockdown and CRISPR knockout of DNMT1 and DNMT3B across a panel of AML cell lines screened in Project Achilles. DepMap categorizes DNMT1 as a common essential gene (a gene that ranks within the topmost depleting genes in at least 90% of cancer cell lines tested) based on the gene effects observed across a large number of cell lines screened using the shRNA and CRISPR libraries[25,26]. This trend is observed when looking at DNMT1 shRNA and CRISPR data across the AML cell line panel (Fig. 1a). Of the AML cell lines screened, OCI-AML2 and OCI-AML3, which are both mutant for DNMT3A, exhibited some of the strongest gene effects in both the CRISPR and RNAi datasets compared to the other AML lines screened, which are wild-type for DNMT3A (Fig. 1a). In contrast, DNMT3B knockdown/knockout did not strongly impact cell proliferation across AML cell lines (Fig. 1b).

We next sought to validate these findings, first by shRNA knockdown of DNMT1 and/or DNMT3B in AML cell lines either wild-type or mutant for DNMT3A. THP-1 cells, which are wild-type for DNMT3A, and OCI-AML2 cells, which harbor a homozygous R635W mutation in the MTase domain of DNMT3A, were transduced with shRNAs targeting DNMT1 and/or DNMT3B. A shRNA targeting luciferase was used as a negative control. QPCR experiments verified efficient knockdown of DNMT1 and DNMT3B transcript levels in THP-1 and OCI-AML2 cells (Fig. 1c). Previous reports suggest that knockdown of DNMT3B expression in a DNMT3A-mutant AML cell line results in increased DNMT1 expression[18]. Although we did not observe an increase in DNMT1 transcript levels with shRNA knockdown of DNMT3B in either cell line, knockdown of DNMT3B expression did lead to a modest decrease in DNMT1 transcript levels in OCI-AML2 cells. Furthermore, shRNA knockdown of DNMT1 did not affect the expression of DNMT3B in THP-1 cells whereas it decreased the expression of DNMT3B in OCI-AML2 cells (Fig. 1c).

Growth of THP-1 and OCI-AML2 cells transduced with shRNAs targeting DNMT1 and/or DNMT3B was assessed using an 8 day Incucyte-based spheroid-like growth assay. While shRNA knockdown of DNMT1 significantly impacted cell growth in both THP-1 and OCI-AML2 cells (Fig. 1d, e and Supplementary Fig. 1), shRNA knockdown of DNMT3B had disparate effects on growth in the cell lines tested. Whereas knockdown of DNMT3B exhibited subtle effects on cell growth in OCI-AML2 cells, DNMT3B shRNAs significantly affected proliferation of THP-1 cells (Fig. 1e–e and Supplementary Fig. 1). Additionally, knockdown of both DNMT1 and DNMT3B had a greater effect on cell growth in OCI-AML2 cells than knockdown of each

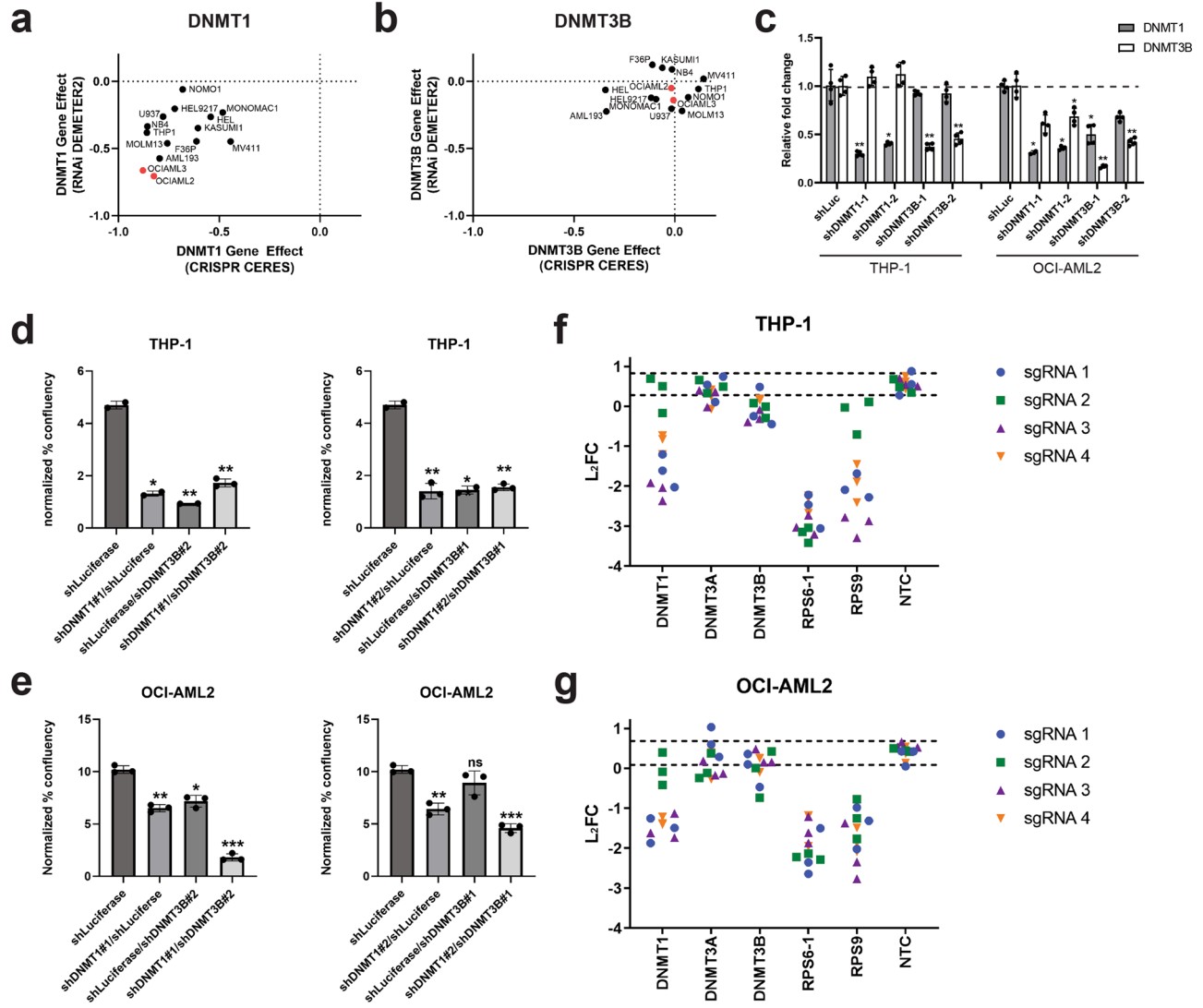

**Fig. 1 Knockdown and knockout of DNMT1 inhibits cell proliferation in AML cell lines independent of the mutational status of *DNMT3A*. a, b** RNAi DEMETER2 and CRISPR CERES scores from publicly available AML cell lines targeting *DNMT1* (**a**) and *DNMT3B* (**b**). The *DNMT3A* mutant AML cell lines, OCI-AML2 and OCI-AML3, are highlighted in red. **c** Relative transcript levels of DNMT1 and DNMT3B in THP-1 and OCI-AML2 cells transduced with DNMT1 and/or DNMT3B shRNAs. Relative transcript levels were normalized to GAPDH and fold change was calculated relative to cells treated with a shRNA targeting luciferase (shLuc). $n = 4$; *$P < 0.05$, **$P < 0.01$, Dunnet's multiple comparison test. **d, e** Spheroid-like growth assays on THP-1 cells, a cell line wild-type for *DNMT3A* (**d**), and OCI-AML2 cells, which is mutant for *DNMT3A* (**e**), transduced with DNMT1 and/or DNMT3B shRNAs. Spheroid-like cell confluence was normalized to the confluence on day 0 of the assay to calculate normalized % confluency. $n = 3$. *$P < 0.01$, **$P < 0.001$, ***$P < 0.0001$, ns = not significant, unpaired t-tests. **f, g** CRISPR/Cas9 pooled screen data for sgRNAs targeting *DNMT1*, *DNMT3A*, and *DNMT3B* in THP-1 (**f**) and OCI-AML2 (**g**) cells. Four sgRNAs targeting each gene are shown, with experiments being performed in triplicate. *RPS6* and *RPS9* are common essential genes used as positive controls that affect viability. NTC are non-targeting control guides that should not affect cell growth. The dashed lines represent the mean $L_2FC +/-2$ standard deviations (S.D.) of the non-targeting control sgRNAs included in the library.

individual gene (Fig. 1e and Supplementary Fig. 1). These data suggest that the level of expression of DNMT family members affects cell growth of AML cell lines.

Effects on proliferation were also tested using a CRISPR/Cas9 pooled screen in THP-1 and OCI-AML2 cells. As observed in shRNA knockdown experiments, sgRNAs targeting *DNMT1* were depleted in both THP-1 and OCI-AML2 cells, suggesting that knockout mutations in *DNMT1* inhibits proliferation. Conversely, sgRNAs targeting either *DNMT3A* or *DNMT3B* did not affect cell proliferation (Fig. 1f, g). These findings are consistent with published CRISPR/Cas9 screens performed in AML cell lines by independent groups[27,28]. Taken together, these data suggest that AML cell lines are sensitive to reduced expression of DNMT1, regardless of the mutational status of *DNMT3A*.

**A CRISPR/Cas9 tiling screen targeting *DNMT1* and *DNMT3B* identified the catalytic MTase domain of *DNMT1* as essential for proliferation in AML cell lines.** CRISPR/Cas9 tiling screens have proven instructive in identifying domains within a given protein that are essential for its function. Previous studies have shown that sgRNAs targeting essential protein domains result in a stronger phenotype than sgRNAs targeting non-essential regions[19–21]. These findings led to the hypothesis that in-frame mutations could have considerable impacts on essential protein domains by disrupting important structural or functional conformations, while those that affect non-essential regions may not have a phenotypic effect.

To identify if any functional domains within DNMT1 or DNMT3B are essential for viability, we performed CRISPR/Cas9

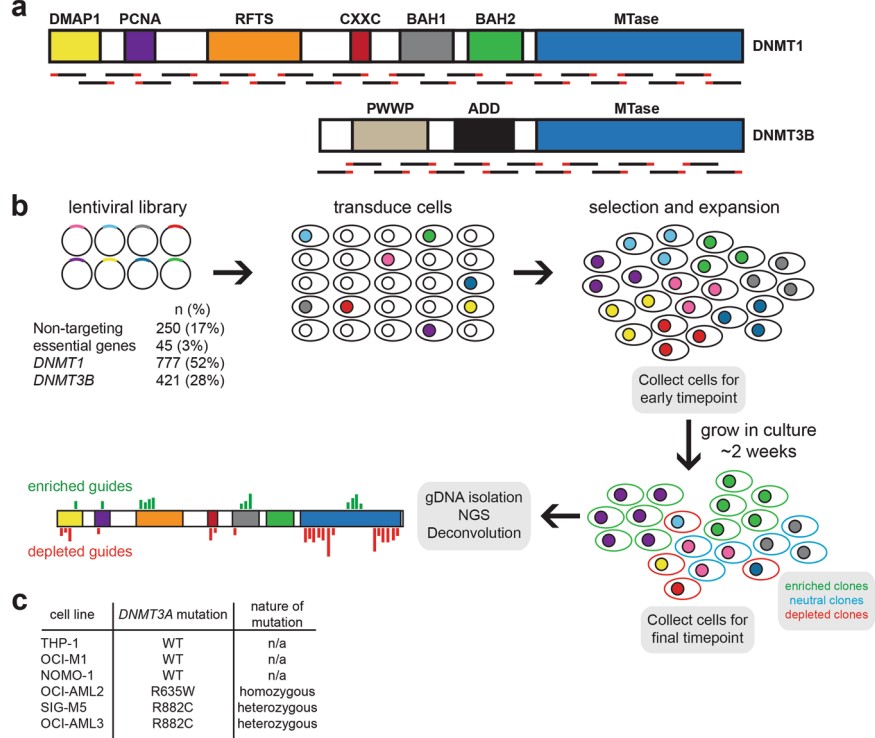

**Fig. 2 Pooled screen design. a** Domain architecture of DNMT1 and DNMT3A proteins. Domain abbreviations: DMAP1, DNMT1-associated protein motif; PCNA, proliferating cell nuclear antigen motif; RFTS, replication focus targeting sequence; CXXC, zinc-finger binding domain; BAH, bromo-adjacent homology domain; PWWP, Pro-Trp-Trp-Pro domain; ADD, ATRX-DNMT3-DNMT3L domain; MTase, methyltransferase domain. sgRNAs were designed to tile the coding regions of both genes. **b** Screening procedure. sgRNAs targeting *DNMT1* and *DNMT3B* were synthesized and constructed as a pooled virus. Non-targeting control sgRNAs and sgRNAs targeting common essential genes were included as negative and positive controls, respectively. Cells were transduced at an MOI between 0.3 and 0.5 and grown on puromycin selection for 3 weeks. Cells were collected 3 days post-puromycin selection and on the last day of the screen. Isolation of genomic DNA and NGS were performed to calculate sgRNA enrichment and depletion over time. sgRNA enrichment and depletion was mapped along the coding regions of *DNMT1* and *DNMT3B* to determine any functional domains essential for viability. **c** AML cell lines chosen for pooled screens. THP-1, OCI-M1, and NOMO-1 cells are wild-type for *DNMT3A*, whereas OCI-AML2, SIG-M5, and OCI-AML3 contain mutations within the MTase domain of *DNMT3A*.

tiling screens in AML cell lines wild-type or mutant for *DNMT3A*. A pooled lentiviral library was generated to target the entire coding regions of both *DNMT1* and *DNMT3B* (Fig. 2a), with non-targeting sgRNAs and sgRNAs targeting essential genes included as negative and positive controls, respectively (Fig. 2b). The pooled lentiviral library was transduced into cells at a low multiplicity of infection (MOI) to ensure that each cell received at most 1 sgRNA. Cells were selected on puromycin and expanded over 3 weeks. Genomic DNA was isolated from the final timepoint and an earlier timepoint (5 days post-transduction) and next generation sequencing (NGS) was performed to identify sgRNAs that depleted over time (Fig. 2b). Six AML cell lines were chosen for screening based on their *DNMT3A* mutational status (Fig. 2c).

We first assessed whether knockout mutations in *DNMT3B* would impact proliferation. Overall, sgRNAs targeting the coding region of *DNMT3B* did not exhibit strong effects on proliferation in *DNMT3A* wild-type cell lines NOMO-1, OCI-M1, and THP-1 (Fig. 3a–c) and *DNMT3A* mutant cell lines OCI-AML2, SIG-M5, and OCI-AML3 (Fig. 3d–f). In contrast, sgRNAs targeting *DNMT1* exhibited different levels of sensitivity across the AML cell lines screened. In AML cell lines wild-type for *DNMT3A*, sgRNAs spanning *DNMT1* exhibited varying degrees of proliferative effects in NOMO-1, OCI-M1, and THP-1 cells (Fig. 4a–c). Similarly, knockout mutations spanning *DNMT1* exhibited varying proliferative effects across *DNMT3A* mutant AML cell lines OCI-AML2, SIG-M5, and OCI-AML3 cells

(Fig. 4d–f). Interestingly, sgRNAs targeting specific functional domains within DNMT1 strongly inhibited cell proliferation across most of the cell lines screened. Specifically, sgRNA depletion was observed in the replication foci targeting sequence (RFTS), bromo adjacent homology 1 (BAH1), BAH2, and the catalytic MTase domains. However, sgRNAs targeting the most N-terminus of DNMT1, containing the DMAP1 and PCNA domains, did not exhibit strong effects on proliferation in any AML cell line screened, despite those sgRNAs being more likely to generate out-of-frame mutations early in the protein sequence. Taken together, these results suggest that *DNMT3B* is not synthetic lethal with *DNMT3A* mutations observed in AML cell lines. Furthermore, CRISPR-generated mutations in certain functional domains within DNMT1 exhibit proliferative disadvantages in AML cell lines wild-type and mutant for *DNMT3A*.

**Statistical analyses identify the MTase domain of DNMT1 as the protein domain most essential for viability in AML cell lines.** Although the *DNMT1/DNMT3B* tiling screens identified certain protein domains within DNMT1 that are essential for proliferation, we observed a range of sensitivity to sgRNAs targeting *DNMT1* across cell lines. Furthermore, sgRNAs targeting some functional domains had minimal effect on proliferation in most of the cell lines screened. To determine whether sgRNAs targeting different DNMT1 functional domains exhibited comparable types of mutations across AML cell lines, we transfected individual sgRNAs into OCI-AML2 (*DNMT3A* mutant) and

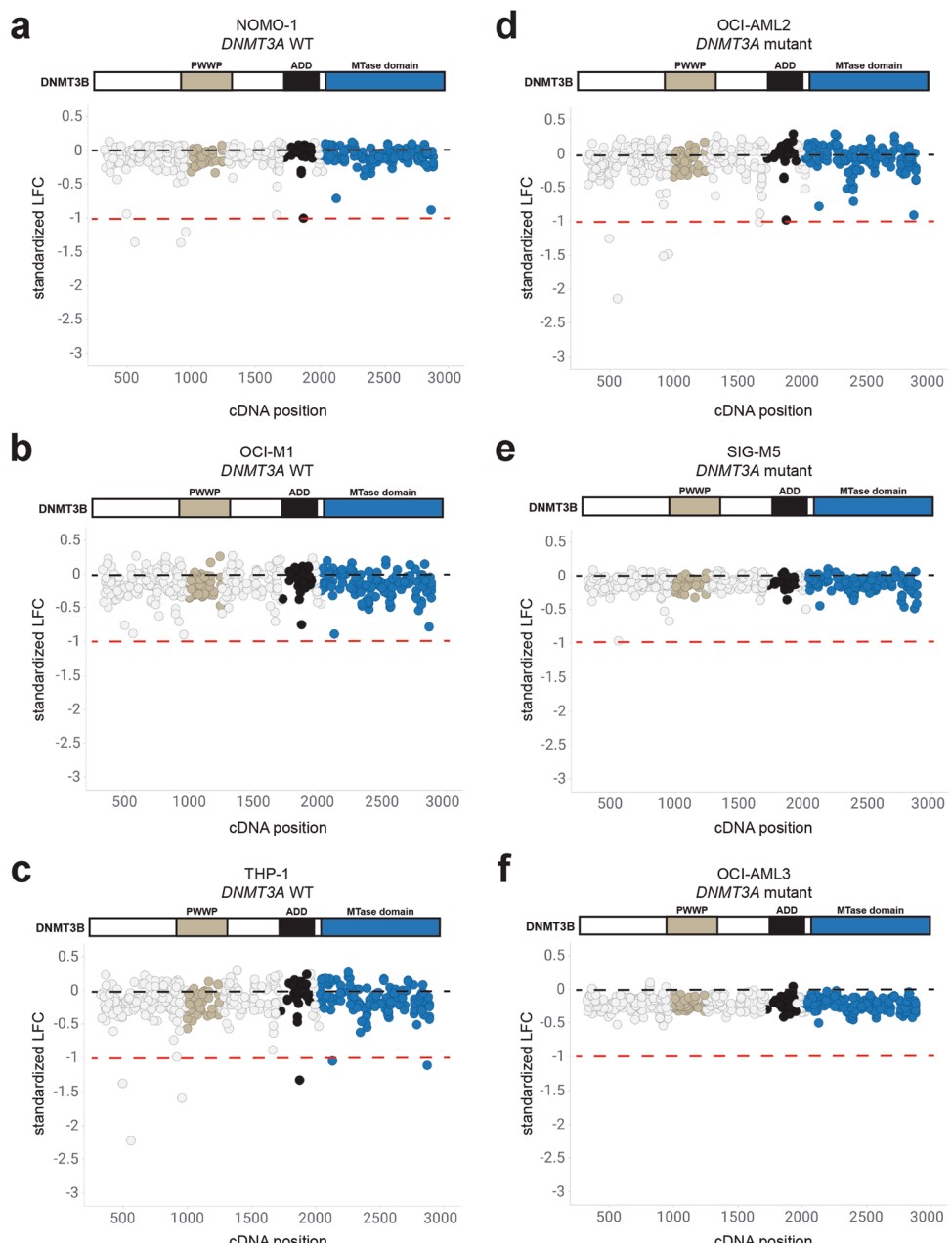

**Fig. 3 Knockout mutations spanning the DNMT3B protein have minimal effects on proliferation across AML cell lines wild-type and mutant for** ***DNMT3A.*** **a–c** sgRNAs tiling *DNMT3B* in AML cell lines wild-type for *DNMT3A*, NOMO-1 (**a**), OCI-M1 (**b**), and THP-1 (**c**) are shown with respect to their cDNA position along the coding region of DNMT3B. **d–f** sgRNAs tiling *DNMT3B* in AML cell lines mutant for *DNMT3A*, OCI-AML2 (**d**), SIG-M5 (**e**), and OCI-AML3 (**f**) are shown with respect to their cDNA position along the coding region of DNMT3B. Each circle represents an individual sgRNA targeting *DNMT3B*. The standardized LFC value for each sgRNA is plotted on each graph. The dashed black bar denotes the mean standardized LFC of the non-targeting controls for each cell line. The mean standardized LFC of the common essential sgRNAs is denoted by the red dashed line. The color of each circle represents the functional domain it targets: PWWP (light gray); ADD (black); MTase domain (blue). *n* = 3 replicates for each cell line screened.

THP-1 cells (*DNMT3A* wild-type), which were very sensitive to *DNMT1* mutations, and OCI-M1 cells (*DNMT3A* wild-type), which were less sensitive to *DNMT1* mutations. We assessed the mutational spectrum across three functional domains within DNMT1 – the CXXC domain, which was not essential for proliferation in any cell line tested, and the RFTS and MTase domains, in which we observed certain sgRNAs inhibiting proliferation in some AML cell lines screened.

As expected, each individual sgRNA that was tested with NGS analyses exhibited different percentages of out-of-frame and in-frame insertions and deletions (indels) (Supplementary Fig. 2).

The trends for individual sgRNAs across different cell lines, however, were comparable. For example, although OCI-M1 cells were not as sensitive to sgRNAs targeting the MTase domain compared to the other cell lines, those sgRNAs exhibited comparable percentages of in-frame and out-of-frame indels when compared to OCI-AML2 and THP-1 cells, which exhibited high sensitivity to the same methyltransferase sgRNAs (Supplementary Fig. 2). Similar trends were observed for sgRNAs targeting the RFTS domain (Supplementary Fig. 2). sgRNAs spanning the CXXC domain did not affect proliferation in any AML cell line tested, despite exhibiting indel trends that were

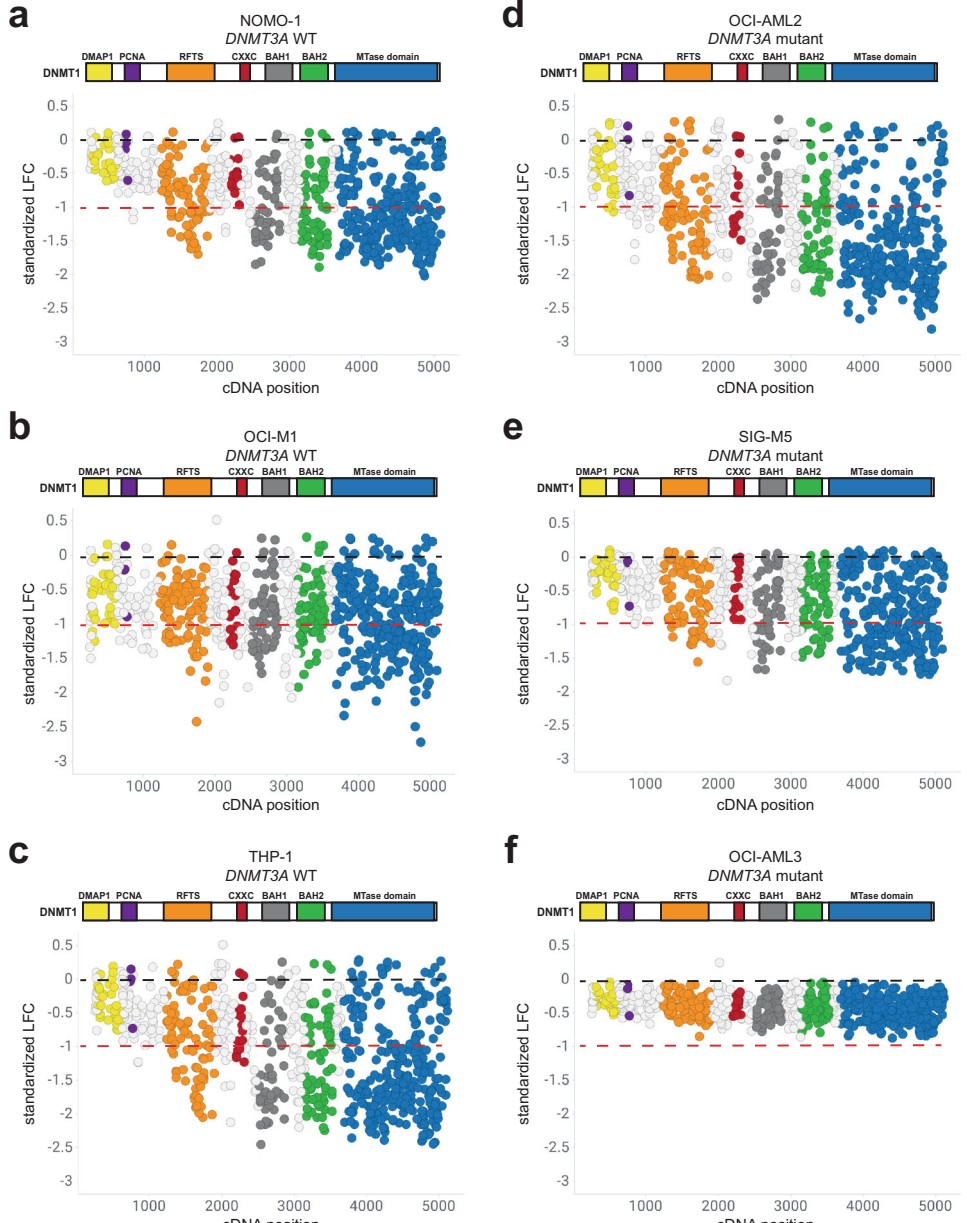

**Fig. 4 Knockout mutations tiling the DNMT1 protein result in proliferative disadvantages across AML cell lines wild-type and mutant for *DNMT3A*. a–c** sgRNAs tiling *DNMT1* in AML cell lines wild-type for *DNMT3A*, NOMO-1 (**a**), OCI-M1 (**b**), and THP-1 (**c**) are shown with respect to their cDNA position along the coding region of DNMT1. **d–f** sgRNAs tiling *DNMT1* in AML cell lines mutant for *DNMT3A*, OCI-AML2 (**d**), SIG-M5 (**e**), and OCI-AML3 (**f**), are shown with respect to their cDNA position along the coding region of DNMT1. Each circle represents an individual sgRNA targeting *DNMT1*. The standardized LFC value for each sgRNA is plotted on each graph. The dashed black bar denotes the mean standardized LFC of the non-targeting controls for each cell line. The mean standardized LFC of the common essential sgRNAs is denoted by the red dashed line. The color of each circle represents the functional domain it targets: DMAP1 (yellow); PCNA (purple); RFTS (orange); CXXC (red); BAH1 (dark gray); BAH2 (green); MTase domain (blue). $n = 3$ replicates for each cell line screened.

comparable to sgRNAs targeting other domains that did exhibit proliferative disadvantages (Supplementary Fig. 2). These analyses suggest that the proliferative effects observed across cell lines were not due to differences in sgRNA effectiveness across cell lines and individual protein domains.

To better assess the essentiality of a functional domain(s) in DNMT1, we utilized 2 different computational approaches. First, we used ProTiler, a computational method to map CRISPR knockout hyper-sensitive (CKHS) regions, which correlates with functional domains associated with a stronger sgRNA depletion effect across the coding region of the protein-of-interest[19].

ProTiler was run on individual sgRNAs spanning either *DNMT1* (Fig. 5a) or *DNMT3B* (Fig. 5b) in AML cell lines wild-type for *DNMT3A*. Several CKHS regions were identified within the DNMT1 protein in NOMO-1 and THP-1 cells, spanning the RFTS, BAH1, BAH2, and MTase domains. One CKHS region within the MTase domain of DNMT1 was also identified in OCI-M1 cells (Fig. 5a). In contrast, very few consistent CKHS regions within *DNMT3B* were identified in any of the *DNMT3A* wildtype AML cell lines screened (Fig. 5b). ProTiler was next run on the AML cell lines mutant for *DNMT3A*. Several CKHS regions within DNMT1 were identified in the OCI-AML2 and OCI-

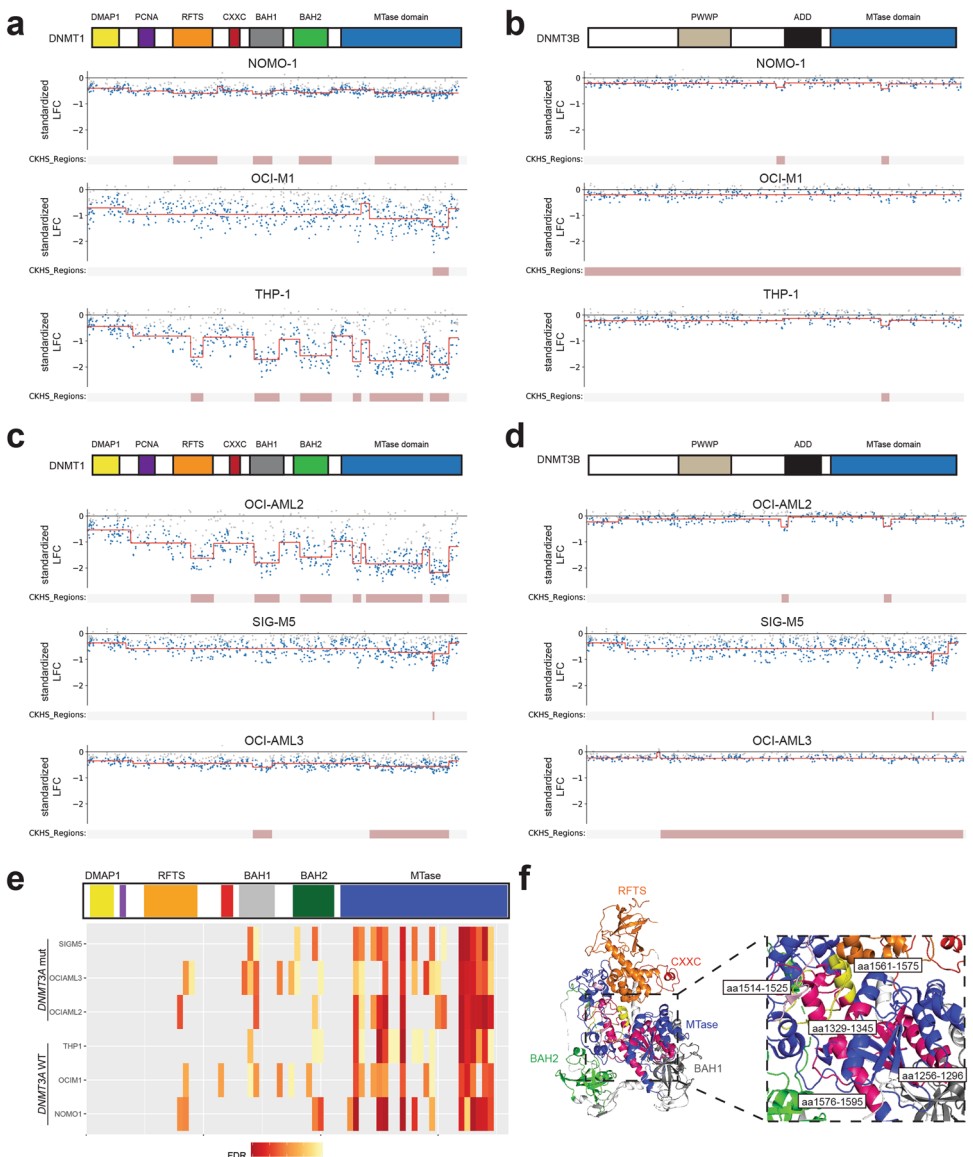

**Fig. 5 Mutations in the catalytic MTase domain of DNMT1 affect proliferation in AML cell lines. a, b** The CKHS profile (from ProTiler) of DNMT1 (**a**) and DNMT3B (**b**) in *DNMT3A* wild-type cell lines NOMO-1, OCI-M1, and THP-1. **c, d** The CKHS profile of DNMT1 (**c**) and DNMT3B (**d**) in *DNMT3A* mutant cell lines OCI-AML2, SIG-M5, and OCI-AML3. **e** STARS analyses on bins of 11 sgRNAs along the span of the DNMT1 protein in SIG-M5, OCI-AML3, OCI-AML2, THP-1, OCI-M1, and NOMO-1 cells. A bin with an FDR of less than 0.25 was considered to have a significant impact on cell proliferation. The PCNA (purple) and CXXC (red) functional domains are not labeled on the DNMT1 protein structure. **f** 3D structure of DNMT1 (amino acids 291-1620; PDB ID: 4WXX) with functional domains RFTS (orange), CXXC (red), BAH1 (gray), BAH2 (green), and MTase (blue) domains labeled. The amino acids spanning the sgRNA bins in the MTase domain that gave a STARS FDR < 0.25 are colored in magenta and labeled on the structure with the respective amino acids position(s) that the sgRNAs targeted. The 2 bins that were the most consistent hits across the cell lines screened screens (aa 1514–1525 and aa 1561–1575) are highlighted in yellow.

AML3 datasets, again covering the BAH1, and MTase domains in both cell lines, and additionally the RFTS and BAH2 domains in OCI-AML2 cells. In the SIG-M5 dataset, one CKHS region was identified within the MTase domain of DNMT1 (Fig. 5c). As expected, ProTiler did not identify many consistent CKHS regions within DNMT3B based on the OCI-AML2, SIG-M5 and OCI-AML3 datasets (Fig. 5d).

Next, we used STARS, a gene-ranking system to generate false-discovery rates (FDR) to identify significantly depleted regions[22]. CRISPR sgRNAs along the span of both *DNMT1* and *DNMT3B* were grouped into bins of 11 neighbors to act as "pseudo-genes" for the purpose of the STARS algorithm. STARS was run across *DNMT1* (Fig. 5e) and *DNMT3B* (Supplementary Data 1) sgRNAs

for all AML screens. Interestingly, the most significant regions with the strongest FDR values were identified within the MTase domain, especially the more C-terminal region of the catalytic domain (Fig. 5e). Mapping of the statistically significant MTase domain bins identified with STARS on the crystal structure of the DNMT1 protein[29] helps to visualize the location of the most dependent regions of the catalytic domain and grasp the extent of the domain that is required for proliferation across the AML cell lines screened (Fig. 5f). The 2 most consistent bins of sgRNAs identified across the majority of screens (covering amino acids 1514–1525 and 1561–1575) target one of the 10 conserved catalytic motifs (motif IX) and the target recognition domain, which recognizes the sequence to be methylated. Taken together,

our data suggest that the MTase domain of DNMT1 is the most essential functional domain in AML cell lines with respect to proliferation.

**Knockout mutations spanning *DNMT1* confer sensitivity in isogenic AML cell line pairs with clinically relevant *DNMT3A* mutations**. Our data across *DNMT3A* wild-type and mutant cell lines does not clearly suggest that *DNMT1* is synthetic lethal with *DNMT3A* mutation. Our shRNA experiments found that AML cell lines are sensitive to *DNMT1* knockdown regardless of *DNMT3A* mutational status. Additionally, published and internal CRISPR genome-wide screens found that both *DNMT3A* wild-type and mutant AML cell lines are sensitive to *DNMT1* knock-out mutations[27,28]. A caveat of these experiments is that in addition to the mutational status of *DNMT3A*, AML cell lines have many differences within their genetic backgrounds, making it difficult to make any conclusions on any potential *DNMT1/DNMT3A* genetic interaction. To better understand the relationship between these two genes, we generated isogenic AML cell line pairs, in which each cell line in a pair is only different with respect to their *DNMT3A* mutational status. A clinically relevant R882C mutation in *DNMT3A* was knocked into one allele in OCI-M1 and NOMO-1 cells, which are normally wild-type for *DNMT3A*. Conversely, we reverted *DNMT3A* mutations in OCI-AML2 and SIG-M5 (R635W homozygous and R882C heterozygous mutations, respectively) cells back to the wild-type sequence (Fig. 6a, b). Sanger sequencing verified successful knock-in of the relevant sequences into the *DNMT3A* locus (Fig. 6c–f).

*DNMT1/3B* tiling screens were first performed on OCI-M1 and NOMO-1 parental cells, which are both wild-type for *DNMT3A*, and their *DNMT3A*[R882C] isogenic pairs. Interestingly, we observed an increased depletion of sgRNAs spanning *DNMT1* in OCI-M1 and NOMO-1 cells harboring the *DNMT3A*[R882C] mutation compared to their parental *DNMT3A* wildtype controls (Fig. 6g–j). This trend was observed stronger in the OCI-M1 screens relative to the NOMO-1 screens. Tiling screens were also performed in OCI-AML2 and SIG-M5 parental cells, which are both mutant for *DNMT3A*, and their *DNMT3A* wildtype isogenic pairs. Less dropout of sgRNAs spanning *DNMT1* was observed in OCI-AML2 cells wildtype for *DNMT3A* compared to the parental *DNMT3A* mutant cell line (Fig. 6k, l). In SIG-M5 cells, dropout of sgRNAs spanning *DNMT1* were observed in both the parental (*DNMT3A* mutant) and isogenic (*DNMT3A* wildtype) screens (Fig. 6m, n). For all isogenic cell lines screened, *DNMT3B* sgRNAs were largely unaffected (Supplementary Fig. 3). Furthermore, *DNMT1/3B* tiling screens using a pooled population of NOMO-1 and SIG-M5 parental cells vs. a clonal expansion of the parental cell line exhibited similar trends (Supplementary Fig. 4). To directly compare the LFC values per sgRNA spanning *DNMT1* in the isogenic cell line pairs, we calculated the differential LFC for each isogenic cell line pair, using the parental cell line as the reference. In both the OCI-M1 and NOMO-1 screens, we identified stronger depletion of several sgRNAs spanning the RFTS, BAH1, BAH2, and MTase domains in the *DNMT3A* mutant cell line in comparison to the isogenic wildtype pair, leading to a more negative differential LFC value (Fig. 7a, b). Conversely, in the OCI-AML2 and SIG-M5 screens, in which the parental line is mutant for *DNMT3A*, we identified some sgRNAs, mostly spanning the MTase domain, with more positive differential LFC values (Fig. 7c, d). STARS analysis was performed on the differential LFC values for each isogenic cell line pair. Interestingly, we identified several bins of sgRNAs spanning the MTase domain that were more significantly depleted in the *DNMT3A* mutant cell line compared to the isogenic wildtype pair (Fig. 7e). Some bins targeting other

functional domains, including the RFTS, BAH1, and BAH2 domains, were also identified as more significantly depleted in the *DNMT3A* mutant versus wildtype cell line. Differential LFC analysis on sgRNAs targeting *DNMT3B*, conversely, showed very minimal differences in proliferative effects between the *DNMT3A* wildtype and mutant isogenic cell lines (Supplementary Fig. 5).

We next performed residual analyses to identify significant differences between the clone(s) and reference parental cell lines by fitting data from the pair to a non-linear function and determining the resulting residual of each sgRNA. Neighboring sgRNAs again were binned in groups of 11 to determine an average residual and significance value along the *DNMT1* gene. A positive residual value represents more lethality in the parental line, and a negative residual value represents more lethality in the clone. In the OCI-M1 and NOMO-1 pairs, we observe several regions along the whole *DNMT1* gene with significant and positive residual values (more lethal in the *DNMT3A* wild-type parental line) occurring in the very end of the MTase domain, as well as early in the protein sequence. However, a larger portion of the MTase domain, and BAH1/BAH2 domains exhibit significant and negative residuals, indicating these regions are more lethal in the *DNMT3A* mutant clone (Fig. 7f, g). In the OCI-AML2 isogenic cell line pair, many regions across the *DNMT1* gene were identified, with significant and positive residual values (more lethal in the *DNMT3A* mutant parental line) occurring in the very end of the MTase domain, as well as early in the protein sequence. Some significant and negative residual values (more lethal in the reverted clones) occur in the BAH domains and early in the MTase domain (Fig. 7h). In the SIG-M5 isogenic cell line pairs, we observed regions of significant and positive residuals across the span of DNMT1, but a mix of positive and negative residuals in the very C-terminus of the MTase domain (Fig. 7i). In addition to a mutation in *DNMT3A*, SIG-M5 cells are also mutant for *TET2* and *ASXL-1*, genes which have been shown to be mutated in AML. Although these results suggest a potential dependency on *DNMT1* mutation that correlated with *DNMT3A* mutational status, the subtlety of these results as well as the inconsistency across multiple isogenic cell lines pairs tested preclude us from basing any conclusions on a dependency for DNMT1 in *DNMT3A*-mutated AML.

## Discussion

CRISPR/Cas9 tiling screens have proven a powerful tool for drug target discovery. Tiling sgRNAs across the coding region of a gene-of-interest allows for the identification of individual functional domains as cancer dependencies and could guide more efficacious drug targets for pharmacological inhibition. The findings from previously published tiling screens led to the hypothesis that in-frame mutations generated in essential functional domains can uncover regional vulnerabilities within proteins-of-interest by disrupting catalytic components or important protein:protein interactions, whereas in-frame mutations in non-essential functional domains would have minimal effect on protein function[19–21]. Using the CRISPR/Cas9 tiling screen methodology, we identified the MTase domain of *DNMT1* as a potential dependency in AML. Mutations within the *DNMT1* methyltransferase resulted in proliferative disadvantages across the AML cell lines screened, and consistently scored regardless of analysis method utilized. These findings are not surprising, as the MTase domain is essential for the catalytic function of DNMT1[30].

While sgRNAs targeting the MTase domain had the most impact on proliferation across AML cell lines screened, sgRNAs targeting other functional domains in DNMT1, specifically the RFTS and BAH domains, also exhibited proliferative dependencies, albeit to a lesser extent. The RFTS domain of DNMT1 is essential for

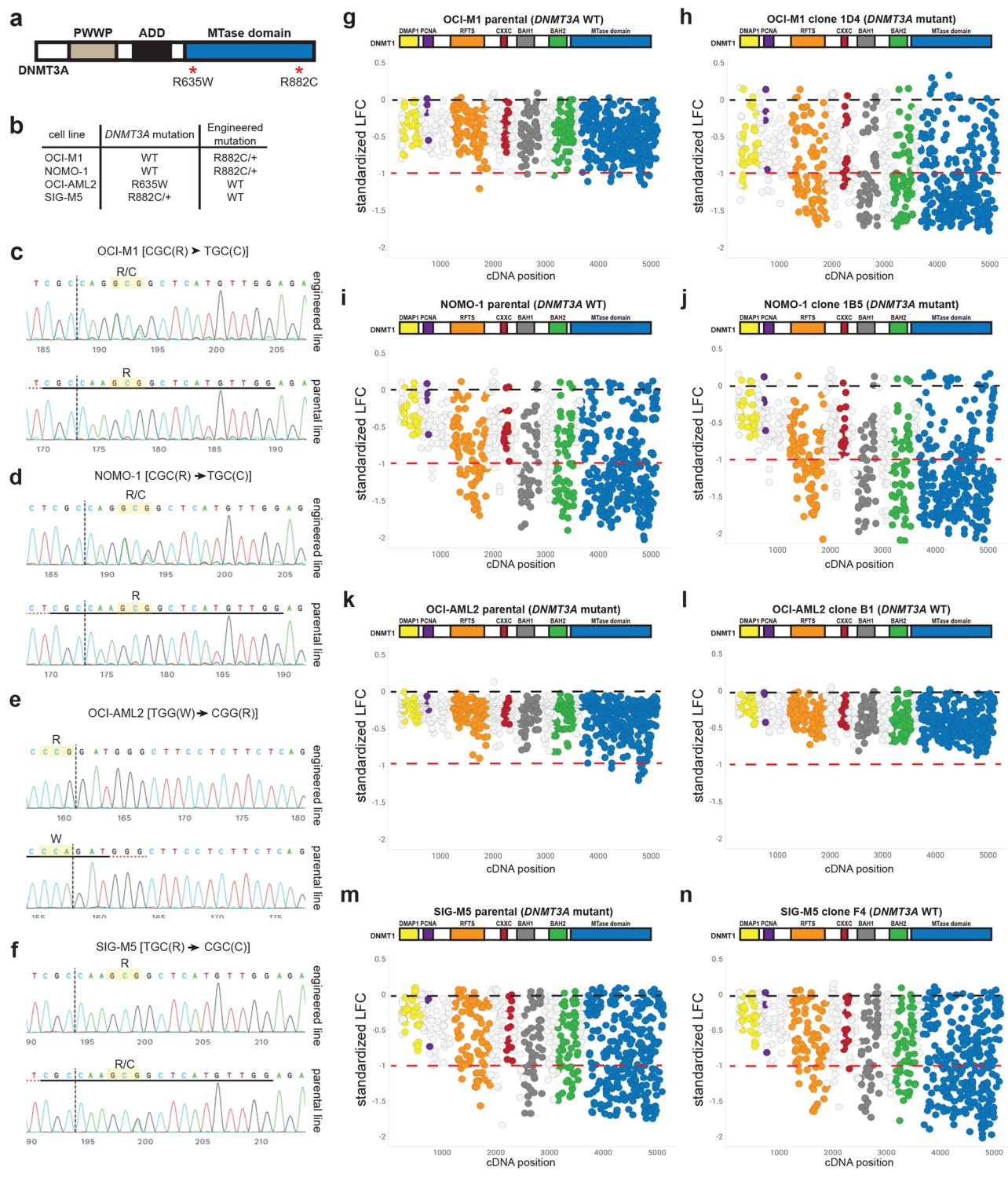

replication-dependent DNA methylation by recruiting DNMT1 to DNA methylation sites[31,32], and can also function as an auto-inhibitory domain of DNMT1[33]. BAH domains can bind to histone tails in a modification-dependent manner and recently, the BAH domains in DNMT1 were reported to be essential for the localization of DNMT1 to replication foci[34]. These domains could be considered as therapeutic approaches in lieu of the MTase domain, as these domains are unique to DNMT1 with respect to catalytically active DNMT family members.

The use of AML cell lines wild-type or mutant for *DNMT3A*, as well as the generation of isogenic *DNMT3A* WT/mutant cell line

pairs, allowed us to better examine the potential synthetic lethal interaction between *DNMT3A* and other DNMT family members *DNMT1* and *DNMT3B*. Although studies have reported that *DNMT3B* is essential for cancer cell growth and survival[35–37], we did not identify an essential role for the de novo methyl-transferase across any AML cell lines with respect to viability in our CRISPR-based screens. Conversely, we did find AML cell lines dependent on *DNMT1* expression at varying degrees. Knockout mutations of *DNMT1* exhibited proliferative effects across all AML cell lines screened independent of the mutational status of *DNMT3A*. However, it is difficult to interpret these data

**Fig. 6 Knockout mutations across the DNMT1 protein exhibit differential sensitivity across isogenic AML cell lines wild-type or mutant for *DNMT3A*.**
**a** Domain architecture of the DNMT3A protein. The R882C/+ and R635W mutations in the MTase domain of *DNMT3A* are labeled. **b** The R882C/+ clinically relevant *DNMT3A* mutation was incorporated into OCI-M1 and NOMO-1 cells, which are normally wild-type for *DNMT3A*. OCI-AML2 cells, which harbor a homozygous R635W mutation in *DNMT3A*, were engineered to express wild-type DNMT3A. SIG-M5 cells, which harbor a heterozygous R882C mutation in *DNMT3A*, were engineered to express wild-type DNMT3A. **c–f** Sanger sequencing of the *DNMT3A* gene verifying the engineered mutation in OCI-M1 (**c**), NOMO-1 (**d**), OCI-AML2 (**e**), and SIG-M5 (**f**) cells. Sanger sequencing of the control parental pool of cells (bottom row), as well as the engineered clone (top row), is shown for each cell line. sgRNAs tiling *DNMT1* are shown with respect to their cDNA position in OCI-M1 parental (*DNMT3A* WT, **g** and OCI-M1 *DNMT3A*[R882C/+] (*DNMT3A* mutant, **h**) cells. **i, j** sgRNAs tiling *DNMT1* are shown with respect to their cDNA position in NOMO-1 parental (*DNMT3A* WT, **i**) and NOMO-1 *DNMT3A*[R882C/+] (*DNMT3A* mutant, **j**) cells. **k, l** sgRNAs tiling *DNMT1* are shown with respect to their cDNA position in OCI-AML2 parental (*DNMT3A* mutant, **k**) and OCI-AML2 *DNMT3A*[W635R] (*DNMT3A* WT, **l**) cells. **m, n** sgRNAs tiling *DNMT1* are shown with respect to their cDNA position in SIG-M5 parental (*DNMT3A* mutant, **m**) and SIG-M5 *DNMT3A*[C882R/+] (*DNMT3A* WT, **n**) cells. Black dashed line denotes the mean standardized LFC of the non-targeting controls. The mean standardized LFC of the common essential sgRNAs is denoted by the red dashed line. The color of each circle represents the functional domain it targets: DMAP1 (yellow); PCNA (purple); RFTS (orange); CXXC (red); BAH1 (dark gray); BAH2 (green); MTase domain (blue). n = 3 replicates for each screen.

due to the variability in genetic background across the AML cell lines screened. *DNMT3A* wildtype/mutant isogenic AML cell line pairs allowed us to study the effects on *DNMT1* knockout mutations in AML cell lines in which the only difference in their genetic background is the mutational status of *DNMT3A*. We found that the isogenic cell line that is mutant for *DNMT3A* sometimes exhibited the strongest proliferative effects with knockout mutations in *DNMT1*. This, in addition to the DepMap CRISPR/shRNA data that shows that *DNMT3A* mutant AML cell lines were the most dependent on *DNMT1* expression[25,26], suggests a potential increased dependency between *DNMT1* and *DNMT3A* that could be utilized as a therapeutic treatment for AML.

The role of DNMT1 in tumorigenesis has been well-studied across several cancer lineages. A conditional knockout of Dnmt1 has been shown to block the development of leukemia, and haploinsufficiency of Dnmt1 delays the progression of leukemogenesis and impair LSC self-renewal[38]. Reduction of *Dnmt1* expression in a murine adrenocortical tumor cell line induces DNA demethylation and inhibits tumorigenesis[39]. In an $Apc^{Min/+}$, $Trp53^{-/-}$ acinar cell pancreatic cancer model, decreased Dnmt1 expression reduces tumor burden, but not tumor size[40]. Reduction of Dnmt1 in an $Apc^{Min/+}$ colorectal cancer mouse model completely suppresses polyp formation and a reduction in hypermethylation[41]. The role of the maintenance methyltransferase has also been well-studied in breast cancer models. Dnmt1 is essential for mammary and cancer stem cell maintenance and tumorigenesis[7]. Furthermore, DNMT1 promotes tumorigenesis in breast stromal fibroblasts, and reduction of DNMT1 expression suppresses these properties[42]. These studies support our findings that DNMT1 is required for cancer cell survival and makes it an interesting therapeutic target.

To date, the most widely used epigenetic modulators as a therapeutic agent are pan DNMT inhibitors 5-azacitidine (azacitidine) and 5-aza-2′-deoxycytidine (decitabine). Azacitidine and decitabine are pyrimidine analogs that are modified in position 5 of the pyrimidine ring. They incorporate into RNA and DNA and covalently trap DNMT1, leading to protein degradation and reduced DNA methylation. Azacitidine and decitabine are approved globally and are treatments for MDS or AML as single agents or in combination[43]. While effective, there remains a need for more potent and selective inhibitors that can improve outcomes for these patients. A few studies have reported the generation of selective DNMT1 and DNMT3B inhibitors[44–46]. The question that remains is whether a therapeutic window exists for *DNMT3A* mutated AML with a selective DNMT1 inhibitor.

## Method
**Cell lines**. The parental cell lines used in this study are the following: OCI-AML2 (ACC 99, DSMZ), OCI-AML3 (ACC 582, DSMZ) SIG-M5 (ACC 468, DSMZ),

THP-1 (ACC 16, DSMZ), NOMO-1 (ACC 542, DSMZ), and OCI-M1 (ACC 529, DSMZ). OCI-M1 and NOMO-1 cells were sent to Synthego (Menlo Park, CA) for knock-in of the clinically relevant *R882C/+* mutation into the *DNMT3A* locus. The *DNMT3A* mutations in OCI-AML2 and SIG-M5 cells were also engineered back to the wild-type sequence by Synthego. Clonal expansions of NOMO-1 and SIG-M5 parental pools obtained from Synthego were generated with a limited dilution series in 384-well plates. All cell lines used in this study were authenticated and confirmed mycoplasma negative at Genetica/LabCorp. Cas9-expressing OCI-M1 cells (OCI-M1_311 cas9) were obtained from the Broad Institute. For the remaining cell lines used in this study, Cas9-expressing cell lines were generated by lentiviral transduction using one of the following: pXPR_BRD101 (Broad Institute), pLEX_311-Cas9v2 (Broad Institute), pLV-ENO-Cas9 (Vector Builder), pLV-TPT1-Cas9 (Vector Builder), pLV-TUBA1B-Cas9 (Vector Builder), or pLV-TMSB4X-Cas9 (Vector Builder), which express Cas9 from either the EFS, EF1a, TPT1, TUBA1B, ENO, or TMSB4X promoters, respectively, as well as the blasticidin resistance gene. Cells were selected for 12–14 days post-selection on blasticidin. Cas9 activity for each Cas9-expressing cell line was tested using pXPR_011 or pRDA_047 (Broad Institute), lentiviral GFP reporters that co-expresses a sgRNA guide targeting GFP[47]. A Cas9 cell line with a reduction in GFP expression by at least 50% 10–12 days post-transduction was considered usable for pooled screens.

**Incucyte-based spheroid-like growth assays**. The Incucyte® Live-Cell Analysis System (Sartorius) is an automated image acquisition system placed in a standard tissue culture incubator with an integrated analysis software that captures and analyzes images of cells over time. THP-1 and OCI-AML2 cells were transduced with shRNA lentivirus of one construct followed by antibiotic selection 24 h post-transduction. Cells stably expressing the shRNA were transduced with the second shRNA lentivirus 1 week later, with antibiotic selection 24 h post-transduction. The following shRNAs were obtained from Sigma Aldrich (MISSION® pLKO.1): sh*DNMT1*#1: TRCN0000364186; sh*DNMT1*#2: TRCN0000232751; sh*DNMT3B*#1: TRCN0000424360; sh*DNMT3B*#2: TRCN0000414235; shControl: SHC007. THP-1 and OCI-AML2 cells transduced with both shRNA constructs were seeded in 200 µL media per well in a 96-well round bottom, ultra-low attachment plates (Costar). Cell seeding numbers were chosen based on growth curves to ensure linear growth throughout the experiment (THP-1: 1,200 cells/well; OCI-AML2: 750 cells/well). Spheroid-like growth was monitored in real-time by live-cell imaging (IncuCyteZOOM, Essenbio, 4× objective) acquiring one image every four hours for 8 days. Confluence (%) as a measure of spheroid-like cell confluence size was determined using an integrated analysis tool (Phase Confluence % tool available in the IncuCyte software). Percent confluence at the beginning of the experiment ($t = 0$) was used as baseline.

**RT-qPCR**. To evaluate the effects on *DNMT1* and *DNMT3B* mRNA expression upon shRNA transduction, RT-qPCR was performed. RNA was isolated from transduced cells 10 days post-transduction using the Qiagen RNeasy Mini kit (Qiagen). cDNA was generated using the Invitrogen SuperScript IV Reverse Transcriptase kit (Thermo Fisher). *DNMT1* and *DNMT3B* TaqMan primers were purchased from Thermo Fisher (TaqMan Gene Expression Assay) and the specific Ct values were normalized to the GusB Ct values (housekeeping gene).

**Lentiviral CRISPR/Cas9 libraries**. The Brunello library (CP0041) was used for genome-wide CRISPR/Cas9 pooled screens (Broad Institute)[22]. This library contains 77,441 sgRNAs total, comprising of approximately 4 sgRNAs per gene and 1000 non-targeting control sgRNAs. A custom pooled *DNMT1/3B* lentiviral library consisting of 1493 sgRNAs was created by the Broad Institute. This library contains 777 sgRNAs spanning the coding region of DNMT1 and 421 sgRNAs targeting the coding region of DNMT3B. Non-targeting sgRNAs (250 sgRNAs) and 45 sgRNAs targeting common essential genes were also included in the library as negative and positive controls, respectively. See Supplementary Data 2 for sgRNA sequences.

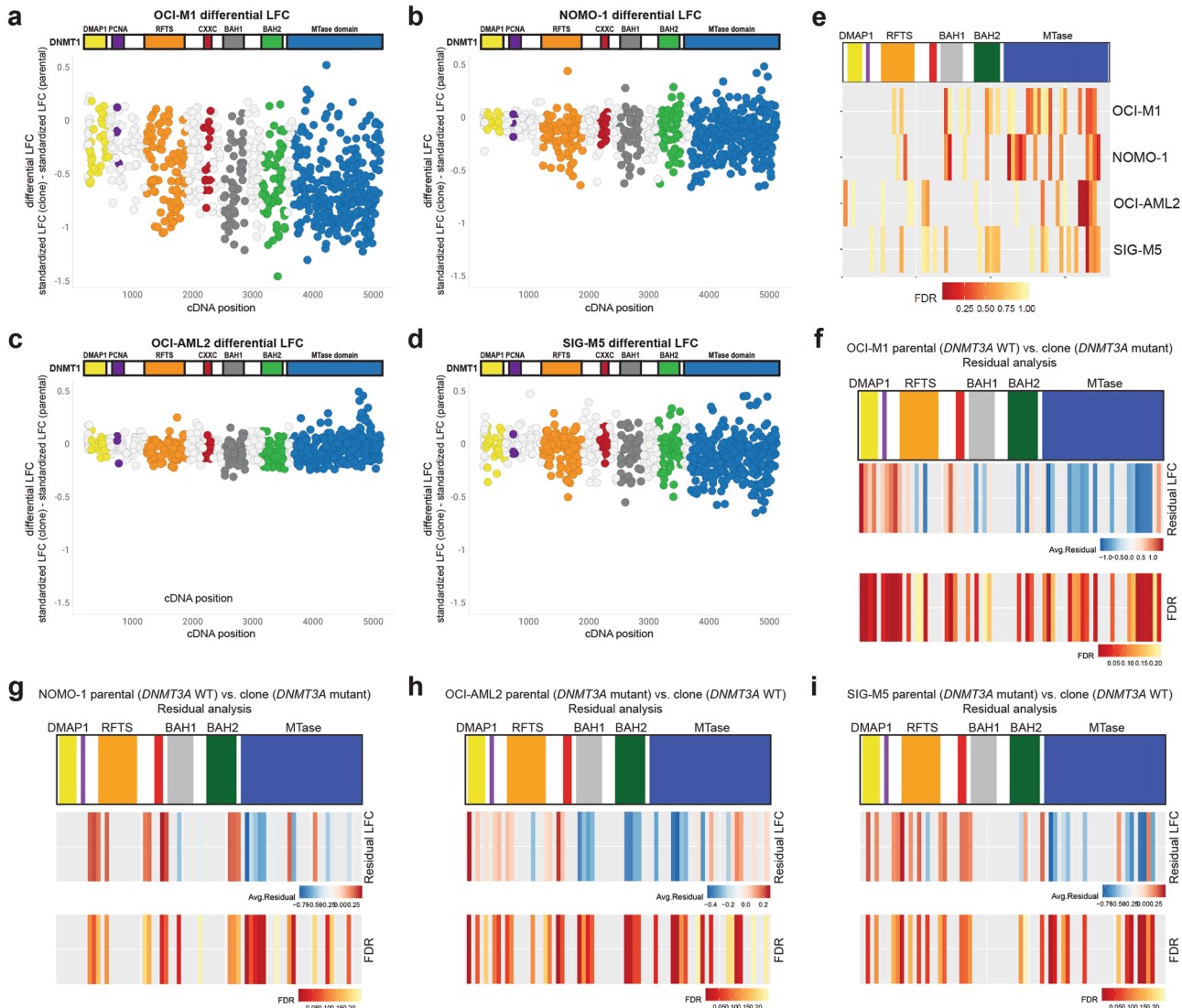

**Fig. 7 Clinically relevant mutations in the MTase domain of DNMT3A confer sensitivity to knockout mutations of *DNMT1* in isogenic AML cell lines wild-type/mutant for *DNMT3A*. a–d** The differential LFC was calculated for each sgRNA targeting DNMT1 across the OCI-M1 (**a**), NOMO-1 (**b**), OCI-AML2 (**c**), and SIG-M5 (**d**) isogenic cell line pairs, using the parental line as a reference. Each circle represents an individual sgRNA spanning the coding region of DNMT1 **e** STARS analyses was performed using bins of 11 sgRNAS along the span of the DNMT1 protein in the OCI-M1, NOMO-1, OCI-AML2, and SIG-M5 isogenic cell line pairs. The differential LFC values for each isogenic cell line pair was used for these analyses. A bin with an FDR of less than 0.25 was considered to have a significant impact on cell proliferation. The PCNA (purple) and CXXC (red) functional domains are not labeled on the DNMT1 protein structure. **f–i** Residual analyses on the OCI-M1 (**f**), NOMO-1 (**g**), OCI-AML2 (**h**), and SIG-M5 (**i**) isogenic cell line pairs. A positive residual (red) indicates more depletion in the parental line compared to the clone, whereas a more negative residual value (blue) indicates more depletion in the clone compared to the parental line. The FDR values across the span of the DNMT1 bins are shown, with an FDR < 0.25 cutoff for significance. The PCNA (purple) and CXXC (red) functional domains are not labeled on the DNMT1 protein structure.

**Pooled screens**. Genome-wide CRISPR pooled screens were performed at a representation of 500× and 3 replicates per screen. *DNMT1/3B* tiling screens were performed at approximately 5000x representation and 3 replicates per screen. Cells were transduced at an MOI between 0.3 and 0.5 to ensure that each cell received at most 1 sgRNA. The volume of virus used to achieve optimal transduction rates was predetermined for each cell line. Cells were spun in 12-well plates at 2000 rpm for 2 h at 30 °C. Cells were grown with puromycin selection beginning at 24 h post-transduction. Cells were passaged every 3–4 days for up to 3 weeks. Cells were collected at days 5 and 23 of the screen, pelleted, resuspended in PBS, and stored at −20 °C. Genomic DNA (gDNA) isolation was performed using the QIAamp DNA Blood Midi Kit or QIAamp DNA Blood Maxi Kit (Qiagen) using manufacturer's instructions. Purified gDNA was sent to the Broad Institute for NGS and deconvolution[22]. Readcounts were normalized to reads per million and log 2 transformed, then a log2 fold change (L2FC) from the plasmid DNA (pDNA) was calculated per sgRNA. Standardized log fold change values were calculated (described below) and used to plot the data as a relative comparison across screens. The raw count values from NGS on all *DNMT1/3B* tiling screens presented in this study are included in Supplementary Data 3.

**STARS, ProTiler, and residual analyses**. STARS analyses[22] were performed for the *DNMT1/3B* tiling screens by taking the log2 fold changes score of each sgRNA. The probability mass function of a binomial distribution was then used to calculate a score for every sgRNA that scores in the top 10% of all sgRNAs for an experiment. Instead of mapping sgRNAs to genes, a mapping file was created for STARS where neighboring sgRNAs were binned in groups of 11 across the coding region of both *DNMT1* and *DNMT3B*. At least 2 sgRNAs in the top 10% were required for a bin of sgRNAs to score, and permutation testing was used to generate the null distribution, allowing for the calculation of p-values and FDRs per bin. The Pro-Tiler algorithm[19] was run on a standardized log2 fold change score, where the median of the non-targeting guides across all screens was set to 0 and the median of the positive 'common essential' controls across all screens was set to −1. The residual analysis[48] uses a natural cubic spline with three degrees of freedom for fitting. The residuals from the fit of the L2FC for the clone(s) compared to the L2FC of the reference parental line, across all sgRNAs in the *DNMT1/DNMT3B* custom library, were averaged across the same bins (instead of by gene) as used for the STARS analysis above. Z-scores p-values were calculated, and FDRs were calculated using the Benjamini & Hochberg method.

**Statistics and reproducibility**. A Dunnet's multiple comparison test was performed on QPCR data shown in Fig. 1b. Unpaired t-tests were performed on Incucyte-based spheroid-like growth assays shown in Fig. 1d, e. Statistical methods performed on CRISPR/Cas9 datasets are described in the Methods section "STARS, ProTiler, and Residual analyses". A simple linear regression model was used to calculate $R^2$ values in Supplementary Fig. 4. Four replicates are shown for QPCR experiments, 3 replicates are shown for Incucyte-based spheroid-like growth assays, and 3 replicates were performed for CRISPR/Cas9 experiments.

**Mutational analysis of sgRNA guides**. Individual sgRNAs targeting either the RFTS, CXXC, BAH1, BAH2, or MTase domains were ordered from IDT (see Supplementary Data 4 for sgRNA sequences). Alt-R CRISPR-Cas9 tracrRNA (1072533) and Alt-R S.p. Cas9 nuclease (1081059) were also obtained by IDT. CRISPR Cas9 RNP transfections were performed according to IDT's protocol using the Neon transfection system 10 μL Kit (Thermo Fisher Scientific). Cells were pelleted for 5 min, 300 × g 48 h post-transfection, and resuspended in 50 μL DirectPCR Cell Lysis Buffer (Viagen Biotech) containing 2 μL Proteinase K (Thermo Fisher Scientific). Lysates were incubated at 68 °C for 15 min, followed by 10 min at 98 °C. Regions of *DNMT1* were PCR amplified and purified using the QIAquick PCR Purification Kit using the manufacturer's instructions (Qiagen). Purified PCR product was sent to Genewiz (South Plainfield, NJ) for Amplicon-EZ Next Generation Sequencing and analyses. The following primers were used for PCR amplification: RFTS-F 5′-GAGCAGCTGTAGGCCAAGTC-3′; RFTS-R 5′-A GGCTACCCCAACTGAACCT-3′; CXXC-F 5′-TGGTGGTGTGATCTTGGC TA-3′; CXXC-R 5′-CTGACCTACCTCCGCTCTTG-3′; BAH1-F 5′-GTGGGGG ACTGTGTCTCTGT-3′; BAH1-R 5′-TGAAAGCTGCATGTCCTCAC-3′; BAH2-F 5′-GGGGGAGTCTACCTTGCAGT-3′; BAH2-R 5′-CGAGGAAGTAGAAGCGG TTG-3′; SAM-H-F 5′-GTTGCAGTGAGCCAAGATCA-3′; SAM-H-R 5′-TGAC GGTTGTGCTGAAGAAG-3′; SAM-L-F 5′-TCACTTGAGCCTCTGGGTCT-3′; SAM-L-R 5′-ATCAGTGCATGTTGGGGATT-3′.

**Reporting summary**. Further information on research design is available in the Nature Research Reporting Summary linked to this article.

## Data availability

The raw data used to generate plots shown in Fig. 1 are available in Supplementary Data 5. The raw count values obtained from the NGS for all *DNMT1/3B* tiling screens presented in this study are available in Supplementary Data 3. There is a restriction on data availability from the CRISPR genome-wide screens shown in Fig. 1. The full dataset is not yet available for publication and we were given approval to show the data for a small subset of genes.

## Code availability

Data analysis was performed using the following previously published algorithms and software: STARS[22], ProTiler[19], and residual analysis[48]. Custom R (version 3.6) code was used for data handling and visualization and is available upon request.

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

## Acknowledgements

We thank Jacqueline Bussolari, Kurtis Bachman, and Yu Sun for critical reading of the manuscript, and Barbara Morschhäuser for helpful discussions on this work.

## Author contributions

B.B., U.P., and G.S.C. conceptualized the study and designed the experiments. B.B., R.C., A.M., and M.C.K. performed the experiments. B.W. and B.B. performed the data analyses for all the CRISPR/Cas9 pooled screens. B.B., B.W., and R.C. wrote the paper with input from all authors.

## Competing interests

All authors were employees at Janssen Research and Development, LLC during this study.
