## [Peer Review File · Communications Biology]

Reviewers' comments:

Reviewer #1 (Remarks to the Author):

Bhogal et al. test the hypothesis that inhibition of paralogs of tumor suppressor genes results in synthetic lethality. In this case, they test if combined disruption of DNMT3B and DNMT1 decreases viability in leukemia cells with DNMT3A mutations, in hopes to potentially use this therapeutically. They survey publicly available RNAi and CRISPR screens and find that 2 DNMT3A-mutated AML cell lines appear somewhat more sensitive to DNMT1 disruption. Yet in their validation experiments using knock-down or CRISPR-mediated disruption of DNMT1 they observe increased drop-out regardless of DNMT3A mutational status, while disruption of DNMT3B has minimal effect. CRISPR tiling screens across DNMT1 and DNMT3B ORFs largely support no difference between DNMT3A genotypes. Finally, by using gene-edited isogenic pairs wherein DNMT3A mutation is corrected to wt in OCI-AML2 cells or DNMT3A R882C mutation is engineered into OCI-M1 cells, they observe somewhat increased drop-out in DNMT1 CRISPR tiling assays.

Overall, this is an interesting hypothesis that is having a really hard time to be validated experimentally. In this case, the claim in the title is barely supported by the data; the authors admit it themselves (see lines 130-131 and 240-241). In addition, the study suffers from occasional data over- and misinterpretation and confusing presentation, as described below.

Figure 1C and lines 111-113: The authors state that "shRNA knockdown of DNMT1 [...] had minimal effects on DNMT3B expression in OCI-AML2 cells" – on the contrary, Fig. 1C demonstrates a significant effect, albeit in the opposite direction compared to a previous report.

Line 115: "spheroid growth assay" – this is confusing. Leukemia cell lines are maintained in suspension, and according to the Methods section were grown in liquid culture. Hence, it is unclear how they would form spheroids. If the algorithm used to analyze image-based cell growth was an algorithm initially developed for spheroids, this should be clarified.

Figure 1D: It is hard to rigorously assess differences in cell growth between the two cell lines as THP-1s grow slower than OCI-AML2 and the groups may not have diverged (note differences in scale).

Lines 122-123: The manuscript text states that "all the DNMT family members can act cooperatively to sustain the DNA methylation function that ultimately regulates cell growth". Here, the authors overinterpreted the data; as presented, the experiments did not look at DNA methylation. The only conclusion that can be rigorously drawn is that the level of expression of DNMT family members affects cell growth; whether the mechanism is DNA methylation-dependent or -independent remains to be determined.

Figures 3 and 4: grouping data by cell line genotype (DNMT3A wt in Fig. 3 and DNMT3A mut in Fig. 4) is confusing and potentially misleading. The main objective is to compare wt and mut, hence they should be presented side-by-side to facilitate such comparison. In addition, L2FC data are not plotted to scale, further confusing the reader. As presented, the sensitivity to DNMT1 disruption does not seem to be much different between wt and mut. Same applies to Figure 5A and C.

Figure 5F: It would be helpful if locations of scoring sgRNAs were mapped/labeled on the crystal structure. Similarly, description of the functional implications of these structural elements would be informative.

In Figure 6, the authors generate isogenic pairs using CRISPR-Cas9 gene editing of DNMT3A, which are then used for CRISPR tiling assays. As written in the main text and the methods section, edited single-cell clones are compared to unedited bulk populations, which is not an appropriately controlled experiment. (Multiple) unedited single-cell clones that went through the same procedure should be used, to control for random variation present in bulk cultures due to genetic drift.

Discussion: The role of DNMT1 in normal and leukemic hematopoiesis has been well described by Trowbridge et al. yet was not cited.

Reviewer #2 (Remarks to the Author):

DNA methylation plays an important role in differentiation of hematopoietic stem cells (HSC). Loss of function mutations of DNMT3A disrupt the balance between self-renewal and differentiation of HSCs. They are the most frequent finding in age-related clonal hematopoiesis. DNMT3A is mutated in a quarter of de-novo AML patients with the hot spot in the R882 codon.

The authors explored the hypothesis that methyltransferases DNMT1 and DNMT3B could be used as a synthetic lethal therapeutic strategy in DNMT3A-mutant AML. They employed tiling CRISPR-Cas9 screens for DNMT1 and DNMT3B genes alongside with nontargeting and essential genes controls to identify critical functional domains. They observed that disruption of several domains in DNMT1 was detrimental to leukemia cell lines both with the wild type and mutant DNMT3A. This was in accordance with published data showing that DNMT1 as an essential gene. To control for the diverse genetic background in AML cell lines, they created two pairs of isogenic cell lines. They introduced a heterozygous R882C mutation in the DNMT3A wild type childhood AML5-derived cell line THP1 and corrected the homozygous R635W mutation in the adult AML4-derived OCI-AML2 cell line. Tiling CRISPR-Cas9 screens for DNMT1 revealed a higher sensitivity to DNMT1 disruption in the isogenic lines with the mutant DNMT3A. The authors conclude that the methyltransferase domain of DNMT1 is the most essential functional domain for survival of DNMT3A-mutant AML. Finally, they raise the question if there is a therapeutic window for selective DNMT1 inhibitors, given that clinically used DNMT inhibitors decitabine and azacytidine inactivate all DNMTs.

The manuscript is well written; the experiments were carefully executed and documented. However, it is unclear why the authors did not create also an isogenic cell line with the wild type DNMT3A from OCI-AML3 cells with a heterozygous hot spot R882C mutation. This would be more clinically relevant than the much less frequent R635W mutation.

Manuscript entitled "The methyltransferase domain of DNMT1 is a dependency in DNMT3A mutated acute myeloid leukemia":

Revised submission COMMSBIO-21-3351A.

We thank both reviewers for their thorough reading of this manuscript. We believe we have addressed all the concerns raised by both reviewers and respond to each in-line with the original comment in order below. We have included the figure that we significantly modified based on our edits to these comments to allow quick reference for the reviewers.

Reviewer #1 (Remarks to the Author):

Bhogal et al. test the hypothesis that inhibition of paralogs of tumor suppressor genes results in synthetic lethality. In this case, they test if combined disruption of DNMT3B and DNMT1 decreases viability in leukemia cells with DNMT3A mutations, in hopes to potentially use this therapeutically. They survey publicly available RNAi and CRISPR screens and find that 2 DNMT3A-mutated AML cell lines appear somewhat more sensitive to DNMT1 disruption. Yet in their validation experiments using knock-down or CRISPR-mediated disruption of DNMT1 they observe increased drop-out regardless of DNMT3A mutational status, while disruption of DNMT3B has minimal effect. CRISPR tiling screens across DNMT1 and DNMT3B ORFs largely support no difference between DNMT3A genotypes. Finally, by using gene-edited isogenic pairs wherein DNMT3A mutation is corrected to wt in OCI-AML2 cells or DNMT3A R882C mutation is engineered into OCI-M1 cells, they observe somewhat increased drop-out in DNMT1 CRISPR tiling assays.

Overall, this is an interesting hypothesis that is having a really hard time to be validated experimentally. In this case, the claim in the title is barely supported by the data; the authors admit it themselves (see lines 130-131 and 240-241). In addition, the study suffers from occasional data over- and misinterpretation and confusing presentation, as described below.

Comment: Figure 1C and lines 111-113: The authors state that “shRNA knockdown of DNMT1 [...] had minimal effects on DNMT3B expression in OCI-AML2 cells” – on the contrary, Fig. 1C demonstrates a significant effect, albeit in the opposite direction compared to a previous report.

Response: The text has been revised to accurately depict the results from this experiment (lines 100-103): “Although we did not observe an increase in *DNMT1* transcript levels with shRNA knockdown of *DNMT3B* in either cell line, knockdown of *DNMT3B* expression did lead to a modest decrease in *DNMT1* transcript levels in OCI-AML2 cells. Furthermore, shRNA knockdown of DNMT1 did not affect expression of *DNMT3B* in THP-1 cells whereas it decreased the expression of *DNMT3B* in OCI-AML2 cells (**Figure 1C**).”

Comment: Line 115: “spheroid growth assay” – this is confusing. Leukemia cell lines are maintained in suspension, and according to the Methods section were grown in liquid culture. Hence, it is unclear how they would form spheroids. If the algorithm used to analyze imagebased cell growth was an algorithm initially developed for spheroids, this should be clarified.

Response: The description of these experiments has been modified in both the Results (line 106) and Methods (beginning of line 512) sections. A description of the analysis tool used to calculate percent confluence has also been added to the Methods section (lines 524-526).

Furthermore, comparable types of experiments have been published using suspension cell lines. Listed below are 2 examples:

Zhang Y, Zhou SY, Yan HZ, Xu DD, Chen HX, Wang XY, Wang X, Liu YT, Zhang L, Wang S, Zhou PJ, Fu WY, Ruan BB, Ma DL, Wang Y, Liu QY, Ren Z, Liu Z, Zhang R, Wang YF. miR-203 inhibits proliferation and self-renewal of leukemia stem cells by targeting survivin and Bmi-1. *Sci Rep.* 2016 Feb 5;6:19995. doi: 10.1038/srep19995. PMID: 26847520; PMCID: PMC4742816.

Aranda-Tavío H, Recio C, Martín-Acosta P, Guerra-Rodríguez M, Brito-Casillas Y, Blanco R, Junco V, León J, Montero JC, Gandullo-Sánchez L, McNaughton-Smith G, Zapata JM, Pandiella A, Amesty A, Estévez-Braun A, Fernández-Pérez L, Guerra B. JKST6, a novel multikinase modulator of the BCRABL1/STAT5 signaling pathway that potentiates direct BCR-ABL1 inhibition and overcomes imatinib resistance in chronic myelogenous leukemia. *Biomed Pharmacother.* 2021 Dec;144:112330. doi: 10.1016/j.biopha.2021.112330. Epub 2021 Oct 19. PMID: 34673425.

Comment: Figure 1D: It is hard to rigorously assess differences in cell growth between the two cell lines as THP-1s grow slower than OCI-AML2 and the groups may not have diverged (note differences in scale).

Response: The reviewer is correct that THP-1 cells grow slower compared to OCI-AML2 cells. AML is a very heterogeneous disorder and the growth rates between commonly used AML cell lines varies greatly. Taking this into account, the calculations and conclusions we generated from these experiments are based on the relative growth rates of each individual cell line. We do not think that extending the experiment in the slower growing THP-1 cells would change our interpretations of these results. We modified Figure 1D-E to help clarify the results from these experiments, showing the normalized % confluency, within each cell line, in the final time point of the experiment relative to the initial time point, including statistical analyses to assess any differences in growth rates in the different samples. Please refer to Figure 1D-E for this modification.

Comment: Lines 122-123: The manuscript text states that “all the DNMT family members can act cooperatively to sustain the DNA methylation function that ultimately regulates cell growth”. Here, the authors overinterpreted the data; as presented, the experiments did not look at DNA

methylation. The only conclusion that can be rigorously drawn is that the level of expression of DNMT family members affects cell growth; whether the mechanism is DNA methylation dependent or -independent remains to be determined.

Response: We agree with the reviewer that we cannot conclude any effects on DNA methylation patterns. We have modified this statement to the following (lines 113-114): “These data suggest that the level of expression of DNMT family members affects cell growth of AML cell lines.”

Comment: Figures 3 and 4: grouping data by cell line genotype (DNMT3A wt in Fig. 3 and DNMT3A mut in Fig. 4) is confusing and potentially misleading. The main objective is to compare wt and mut, hence they should be presented side-by-side to facilitate such comparison. In addition, L2FC data are not plotted to scale, further confusing the reader. As presented, the sensitivity to DNMT1 disruption does not seem to be much different between wt and mut. Same applies to Figure 5A and C.

Response: We addressed this concern in a couple of ways. Every screen we performed had a diverse range of dropout, which is due to several factors, including Cas9 activity of the Cas9-expressing AML cell line and double time of the cell line being screened. Because of this, we decided to plot all screen data using standardized LFC values, which is described in the Methods section (lines 567-570). Please see Figures 3-7 and Supplemental Figures 3-5. This allows an easier side-by-side comparison of the screens performed in this study. Secondly, all *DNMT3B* screen data was moved to Figure 3, and all *DNMT1* screen data was moved to Figure 4. This allows for an easier comparison between *DNMT3A* WT/mutant cell lines as suggested by the reviewer.

Comment: Figure 5F: It would be helpful if locations of scoring sgRNAs were mapped/labeled on the crystal structure. Similarly, description of the functional implications of these structural elements would be informative.

Response: The bins of sgRNAs that were identified as significantly impacted were highlighted in magenta in the original submission. We have made the additional modifications: (1) The 2 bins in the MTase domain that were consistently identified as significant were highlighted in yellow; (2) The amino acid ranges of the significant MTase regions identified and highlighted are listed on the crystal structure. We included a brief description of what is known about the function of the catalytic MTase domain for the 2 most significantly impactful bins of sgRNAs (see line 215-217), mainly that they cover one of the conserved catalytic motifs (motif IX) and the target recognition domain, which recognizes the sequence to be methylated.

Comment: In Figure 6, the authors generate isogenic pairs using CRISPR-Cas9 gene editing of DNMT3A, which are then used for CRISPR tiling assays. As written in the main text and the

methods section, edited single-cell clones are compared to unedited bulk populations, which is not an appropriately controlled experiment. (Multiple) unedited single-cell clones that went through the same procedure should be used, to control for random variation present in bulk cultures due to genetic drift.

Response: To address the concern of reviewer #1 (and reviewer #2), 2 additional isogenic cell line pairs were screened and included in Figures 6-7 and Supplemental Figure 4: NOMO-1 (normally WT for *DNMT3A*) and SIG-M5 (harbors a *DNMT3A*[R882C/+]) mutation). Parental pools and 2 x parental clones of each were screened as controls for these isogenic lines, as suggested by the reviewers. The data is shown in Figures 6-7 and Supplemental Figure 4. We did not observe significant differences in the results between the screens using the pooled parental lines vs. the 2 x individual clones for each isogenic pair screened. Therefore, there are no significant differences to our conclusions, when we perform the analysis using the parental clones rather than the polyclonal parental pool.

Comment: Discussion: The role of DNMT1 in normal and leukemic hematopoiesis has been well described by Trowbridge et al. yet was not cited.

Response: We thank the reviewer for highlighting this omission, and this reference has been added to the discussion in lines 329-331.

Reviewer #2 (Remarks to the Author):

DNA methylation plays an important role in differentiation of hematopoietic stem cells (HSC). Loss of function mutations of DNMT3A disrupt the balance between self-renewal and differentiation of HSCs. They are the most frequent finding in age-related clonal hematopoiesis. DNMT3A is mutated in a quarter of de-novo AML patients with the hot spot in the R882 codon. The authors explored the hypothesis that methyltransferases DNMT1 and DNMT3B could be used as a synthetic lethal therapeutic strategy in DNMT3A-mutant AML. They employed tiling CRISPR-Cas9 screens for DNMT1 and DNMT3B genes alongside with nontargeting and essential genes controls to identify critical functional domains. They observed that disruption of several domains in DNMT1 was detrimental to leukemia cell lines both with the wild type and mutant DNMT3A. This was in accordance with published data showing that DNMT1 as an essential gene. To control for the diverse genetic background in AML cell lines, they created two pairs of isogenic cell lines. They introduced a heterozygous R882C mutation in the DNMT3A wild type childhood AML5-derived cell line THP1 and corrected the homozygous R635W mutation in the adult AML4-derived OCI-AML2 cell line. Tiling CRISPR-Cas9 screens for DNMT1 revealed a higher sensitivity to DNMT1 disruption in the isogenic lines with the mutant DNMT3A. The

authors conclude that the methyltransferase domain of DNMT1 is the most essential functional domain for survival of DNMT3A-mutant AML. Finally, they raise the question if there is a therapeutic window for selective DNMT1 inhibitors, given that clinically used DNMT inhibitors decitabine and azacytidine inactivate all DNMTs.

The manuscript is well written; the experiments were carefully executed and documented. However, it is unclear why the authors did not create also an isogenic cell line with the wild type DNMT3A from OCI-AML3 cells with a heterozygous hot spot R882C mutation. This would be more clinically relevant than the much less frequent R635W mutation.

Response: To address the concern of reviewer #2, we screened 2 x additional isogenic cell line pairs which have been included in Figures 6-7 and Supplemental Figure 4: NOMO-1 (normally WT for *DNMT3A*) and SIG-M5 (harbors DNMT3A[R882C/+] mutation). Because of technical challenges we have experienced with OCI-AML3 using pooled screened approaches (mainly low Cas9 activity leading to low signal in pooled screens), we decided to use SIG-M5 instead. One of the caveats of using SIG-M5 is that in addition to *DNMT3A* mutation, SIG-M5 cells also harbor *TET2* and *ASXL-1* mutations, which are also mutations observed in AML. However, this was the only R882C mutant cell line available to us that was usable for CRISPR/Cas9 screening methods.

Major changes to Figures:

Figure 1: Incucyte-based spheroid-like growth assay figures (Fig. 1D-E below) have been modified from original version to allow for clearer visualization of results and to assess statistical differences in different experimental groups.

Figure 1

Figures 3-4: To allow for easier comparison of *DNMT3A* wildtype and mutant cell lines, all of the *DNMT3B* tiling data was moved to Figure 3, and all of the *DNMT1* tiling data was moved to Figure 4.

Figure 3

Figure 4

Figures 6-7: Data in original Figure 6 was split across 2 new figures (Figures 6-7) to allow for the inclusion of 2 more isogenic cell line pairs.

Figure 6

Figure 7

Supplemental Figure 4: Includes a comparison of results obtained from parental lines using a bulk population of unedited cells and 2 x clonal expansions from the same cell lines.

Supplemental Figure 4

Reviewers' comments:

Reviewer #1 (Remarks to the Author):

Bhogal et al present a revised manuscript wherein they addressed most of the minor points raised by the reviewers. Yet the major problem, that their results contradict what is claimed in the title and the abstract, remains unresolved. This can be rectified by changing the title to more accurately reflect their findings ("The methyltransferase domain of DNMT1 is NOT a dependency in DNMT3A mutated acute myeloid leukemia") and rewriting the abstract accordingly. The authors rightfully admit this on multiple occasions in the manuscript text. OCI-M1 is the only cell line where the main claim of DNMT1 disruption exacerbating DNMT3A^{mut} proliferation actually works, but not in other contexts. Hence, this result is NOT generalizable. I am sorry to be negative here, but keeping the same manuscript title is misleading.

Minor comments:

Figure 1D – what's the difference between right and left?

Line 106 – explain what Incucyte is as not all readers are familiar with the instrument and what it does.

Figures 3-4 – please annotate DNMT3A WT and MUT in the figure to make it easier on the reader.

Figure 5E – there doesn't seem to be a difference between DNMT3A wt/mut genotypes

Lines 266-268 – "These data suggest that the presence of a clinically relevant DNMT3A mutation sensitizes cells to DNMT1 mutations" – you just spent the whole paper showing us this is only occasionally true. Please tone down or remove altogether.

Based on the experimental evidence presented, it is a good idea to replace "generally" by "sometimes" in most instances. Similarly, in line 330, please remove the "strongly" when discussing a questionable synthetic lethal interaction.

Line 349 – [AZA and DAC] "...disrupt the interaction between DNA and DNMTs" – it would be valuable to mention the exact mechanism for a more nuanced understanding (covalent "stapling" of DNMTs to the modified base).

Reviewer #2 (Remarks to the Author):

The major claims of the manuscript are that the methyltransferase domain of DNMT1 is a dependency in DNMT3A mutated acute myeloid leukemia (title) as tiling screens performed in isogenic cell lines showed increased sensitivity to DNMT1 mutations with clinically relevant DNMT3A mutations (abstract).

Yet, this is convincingly shown only in isogenic OCI-M1 cell lines (DNMT3A wt and CRISPR-introduced R882C/+ mutation) and somewhat in OCI-AML2. There was no difference in isogenic pairs or NOMO-1 and SIG-M5. OCI-AML3 cell line showed minimal changes after CRISPR screening, perhaps due to suboptimal expression of Cas9. The manuscript is interesting; however, the authors should modify the title and conclusions to interpret their experimental data more objectively.

Minor points:

The labels in Fig. 1 C and D show identical shRNAs in all experiments (shDNMT1#1 and DNMT3B#2). This is probably an error, since the labels in the original version and in the Supplemental Figure 1 show two different shRNAs for DNMT1 and DNMT3B, respectively

Manuscript entitled " The methyltransferase domain of DNMT1 is an essential domain in acute myeloid leukemia independent of *DNMT3A* mutation":

Revised submission: COMMSBIO-21-3351B

We once again thank both reviewers for their time in reviewing our manuscript. We believe that we have addressed the major concerns raised by both reviewers as well as the minor comments. Below, please find a detailed response to the comments.

Reviewer #1 (Remarks to the Author):

Bhogal et al present a revised manuscript wherein they addressed most of the minor points raised by the reviewers. Yet the major problem, that their results contradict what is claimed in the title and the abstract, remains unresolved. This can be rectified by changing the title to more accurately reflect their findings (“The methyltransferase domain of DNMT1 is NOT a dependency in DNMT3A mutated acute myeloid leukemia”) and rewriting the abstract accordingly. The authors rightfully admit this on multiple occasions in the manuscript text. OCI-M1 is the only cell line where the main claim of DNMT1 disruption exacerbating DNMT3A mut proliferation actually works, but not in other contexts. Hence, this result is NOT generalizable. I am sorry to be negative here, but keeping the same manuscript title is misleading.

Response to reviewer #1: We have revised the title of the manuscript as well as the abstract to address the major concerns that reviewer #1 expressed. Additionally, conclusions made in the manuscript have been modified to reflect the data more objectively. We have highlighted these revised statements within the manuscript document for ease of finding.

Minor comments:

1. Figure 1D – what’s the difference between right and left?

Response: Thank you for catching this error. The 2 graphs for Figure 1D and 1E show results from 2 different shRNA combinations. The x-axis for the graphs on the right in Figures 1D-E have been appropriately corrected. The updated figures have been included at the very end of this document.

2. Line 106 – explain what Incucyte is as not all readers are familiar with the instrument and what it does.

Response: A description of the Incucyte was included in the “Methods” section under “Incucyte-based spheroid-like growth assays,” beginning at line 514: “The Incucyte® Live-Cell Analysis System (Sartorius) is an automated image acquisition system placed in a standard tissue culture incubator with an integrated analysis software that captures and analyzes images of cells over time.”

3. Figures 3-4 – please annotate DNMT3A WT and MUT in the figure to make it easier on the reader.

Response: Cell lines WT or mutant for *DNMT3A* have been annotated in both Figures 3 and 4.

4. Figure 5E – there doesn’t seem to be a difference between DNMT3A wt/mut genotypes

Response: We agree with reviewer #1 on their comment regarding Figure 5E, and do not make any conclusions from Figure 5E claiming that there is a difference between DNMT3A WT/mut genotypes. We addressed this in line 225: “Our data across *DNMT3A* wild-type and mutant cell lines does not clearly suggest that *DNMT1* is synthetic lethal with *DNMT3A* mutation.”

5. Lines 266-268 – “These data suggest that the presence of a clinically relevant DNMT3A mutation sensitizes cells to DNMT1 mutations” – you just spent the whole paper showing us this is only occasionally true. Please tone down or remove altogether.

Response: This sentence was removed from the Results section. Furthermore, we revised the conclusions at the end of the Results section (lines 287-291):

“Although these results suggest a potential dependency on *DNMT1* mutation that correlated with *DNMT3A* mutational status, the subtlety of these results as well as the inconsistency across multiple isogenic cell lines pairs tested preclude us from basing any conclusions on a dependency for DNMT1 in *DNMT3A*-mutated AML.”

6. Based on the experimental evidence presented, it is a good idea to replace “generally” by “sometimes” in most instances. Similarly, in line 330, please remove the “strongly” when discussing a questionable synthetic lethal interaction.

Response: “generally” was removed from both locations that word was used, lines 73 and 331. Additionally, “strongly” was removed from line 332 in the sentence that reviewer #1 referenced.

7. Line 349 – [AZA and DAC] “...disrupt the interaction between DNA and DNMTs” – it would be valuable to mention the exact mechanism for a more nuanced understanding (covalent “stapling” of DNMTs to the modified base).

Response: We modified the text in the discussion to include a description of the mechanism of action on line 350:

“[Aza and dac] incorporate into RNA and DNA and covalently trap DNMT1, leading to protein degradation and reduced DNA methylation.”

Reviewer #2 (Remarks to the Author):

The major claims of the manuscript are that the methyltransferase domain of DNMT1 is a dependency in DNMT3A mutated acute myeloid leukemia (title) as tiling screens performed in isogenic cell lines showed increased sensitivity to DNMT1 mutations with clinically relevant DNMT3A mutations (abstract).

Yet, this is convincingly shown only in isogenic OCI-M1 cell lines (DNMT3A wt and CRISPR-introduced R882C/+ mutation) and somewhat in OCI-AML2. There was no difference in isogenic pairs or NOMO-1 and SIG-M5. OCI-AML3 cell line showed minimal changes after CRISPR screening, perhaps due to suboptimal expression of Cas9. The manuscript is interesting; however, the authors should modify the title and conclusions to interpret their experimental data more objectively.

Response to reviewer #2: We have revised the title of the manuscript as well as the abstract to address the major concerns that reviewer #2 expressed. Additionally, conclusions made in the manuscript have been modified to reflect the data more objectively. We have highlighted these revised statements within the manuscript document for ease of finding.

Minor points:

The labels in Fig. 1 C and D show identical shRNAs in all experiments (shDNMT1#1 and DNMT3B#2). This is probably an error, since the labels in the original version and in the Supplemental Figure 1 show two different shRNAs for DNMT1 and DNMT3B, respectively

Response: Thank you for catching this error. Indeed, the 2 graphs for Figure 1D and 1E show results from 2 different shRNA combinations. The x-axis for the graphs on the right in Figures 1D-E have been appropriately corrected. The updated figures have been included at the very end of this document.

Figure 1D-E: Updated x-axis on the right graphs of Figure 1D and 1E to appropriately show a different combination of shRNAs tested.